# Physiological Correlates of Hypnotizability: Hypnotic Behaviour and Prognostic Role in Medicine

**DOI:** 10.3390/brainsci13121632

**Published:** 2023-11-24

**Authors:** Eleonora Malloggi, Enrica L. Santarcangelo

**Affiliations:** 1Department Translational Research and New Technologies in Medicine and Surgery, University of Pisa, 56127 Pisa, Italy; eleonora.malloggi@med.unipi.it; 2Department of Physics, University of Trento, 38122 Trento, Italy

**Keywords:** motor imagery, functional equivalence, interoception, cerebral blood flow, endothelial function, hypnosis

## Abstract

Studies in the field of experimental hypnosis highlighted the role of hypnotizability in the physiological variability of the general population. It is associated, in fact, with a few differences which are observable in the ordinary state of consciousness and in the absence of suggestions. The aim of the present scoping review is summarizing them and indicate their relevance to the neural mechanisms of hypnosis and to the prognosis and treatment of a few medical conditions. Individuals with high, medium and low hypnotizability scores display different cerebral functional differences—i.e., functional equivalence between imagery and perception/action, excitability of the motor cortex, interoceptive accuracy—possibly related to brain structural and functional characteristics, and different control of blood supply at peripheral and cerebral level, likely due to different availability of endothelial nitric oxide. These differences are reviewed to support the idea of their participation in hypnotic behaviour and to indicate their prognostic and therapeutic usefulness in a few medical conditions.

## 1. Introduction

Hypnotizability, an individual trait substantially stable throughout life [1], is associated with the proneness to experience hypnosis and/or alteration of perception, memory and behavior following the administration of specific suggestions [2]. The observation that suggestions are effective in both the ordinary state of consciousness and after hypnotic induction [3,4,5,6,7,8] has highlighted the role the trait of hypnotizability has, together with other individual traits and with contextual factors, in hypnotic behavior. Hypnotizability, in fact, also predicts the response to suggestions, as placebo and hypnotizability-related mechanisms can cooperate, for instance, in the cognitive control of pain [9]. 

The contribution of several factors to hypnotic behavior is in line with the bio-psycho-social model of hypnosis [10], rather than with the relevance of the induction of the hypnotic state. Standard hypnotizability scales can be used to classify the general population into highly (highs), medium (mediums) and low hypnotizable persons (lows) according to the scales total score or based on the specific scales items the subjects pass [11]. 

Hypnotizability-related physiological correlates are physiological differences associated with different levels of hypnotizability which can be observed in the ordinary state of consciousness and in the absence of suggestions. Amongst others, differences have been found in brain morpho-functional characteristics [12,13], in the functional equivalence between imagery and perception/action, within groups topological homogeneity and modes of information processing [14], excitability of the motor cortex [15,16], vascular peripheral [17,18] and cerebral control of blood flow [19,20], postural and visuomotor control [21,22], interoception [23,24,25], and polymorphism of µ1 receptors [26]. 

The object of this review is the description of the physiological correlates of hypnotizability which can account for a few hypnotic behaviors, or exhibit a prognostic role in medical conditions, or allow personalized pharmacological treatments. From this perspective, we describe facts and the hypotheses which can be suggested based on current evidence.

## 2. Evidence and Related Hypotheses

### 2.1. Cerebral Morpho-Functional and Vascular Correlates of Hypnotizability

The earliest neuroimaging study conducted in individuals with different ability to accept suggestions revealed a larger anterior part of the corpus callosum [27], but recent investigation has not confirmed this difference between hypnotizability groups [28]. Reduction in the entire brain volume (including white and grey matter) has been observed in individuals able to experience hypnosis [29], i.e., the highs. This suggests than not only possible genetic markers, usually associated with localized variations—as occurs, for instance, in schizophrenia [30] and Parkinson disease [31], but also maturation processes related to different availability of maturation factors, i.e., endothelial nitric oxide (NO), could be involved in the observed hypnotizability-related brain volume variations. Other investigations on highs revealed that they display reduced grey matter volume (GMV) in the insula (Figure 1A), larger GMV in the mid-temporal and mid-occipital cortices, stronger functional connectivity between the anterior cingulate and the prefrontal dorsolateral cortex [12]. Cerebellar morpho-functional differences have also been reported (Figure 1A). They consist of reduced GMV in the highs’ left lobules IV–VI compared to lows [13].

The morphological variants of the cerebellar and insula GMV do not indicate impaired functions. Classical cerebellar tasks, such as postural and visuomotor control, in fact, are appropriately although less precisely performed by highs [21,22]. The highs’ postural control is less close than lows’, in fact, the set point for postural control which integrates peripheral sensory reafferents when the body centre of pressure are farther from the origin of sway compared to lows [21]. This means that highs exhibit larger and faster body sway compared to lows, although reporting the same perception of body sway. Nonetheless, they do not fall down and, when they stand up on a very unstable platform, the differences disappear, likely due to attentional effort [32]. Also, visuomotor control, which is modulated by the direction of gaze (during application of prisms and after their removal), is less precise in highs than in lows and both the error and the variability of the error are larger than in lows [22]. In both cases—postural and visuomotor control—the absence of learning across trials characterizes highs. Nonetheless, the typical cerebellar operation—changing the direction of launches toward a target as a function of the application and removal of prisms is appropriately performed. Moreover, the performance of another typical cerebellar test, mental rotation, does not differ between highs, mediums and lows [33].

Hypnotizability-related differences in cerebral blood flow were studied through near-infared spectroscopy (NIRS). It was shown that only highs exhibit a significant increase in blood supply during cognitive tasks (Figure 1B), suggesting that they can better adjust brain oxygenation to metabolic demands [19], which largely depends on both endothelial and neuronal NO release [34]. The metabolic demand, however, could be lower in medium-to-high than in low-to-medium hypnotizables, as suggested by the negative correlation between hypnotizability and cerebrovascular reactivity observed during visual stimulation [20]. This could be due to the highs’ peculiar mode of information processing showing small and distributed network activation changes in the brain [14]. The highs’ greater increase in cerebral blood oxygenation during cognitive tasks could at least partially account for their greater attentional stability depending on the brain dopamine levels (for review, [26]) and, maybe, on cerebellar function (see Section 2.4).

### 2.2. Functional Equivalence between Real and Imagined Perception/Action

The functional equivalence (FE) between actual and imagined perception/action is indicated by the degree of superimposition between the cortical activations observed during these conditions [35,36,37,38,39]. It has been studied in highs and lows through topological analysis of the EEG, which revealed stronger FE between actual and imagined sensori-motor conditions in highs than in lows [14]. In contrast, significant hypnotizability-related differences in the vividness of imagery have not been unanimously reported [40]. EEG topological analysis confirmed the hypothesis based on a behavioral experiment in which the earliest component of the vestibulo-spinal reflex (VR) was elicited by galvanic stimulation of the labyrinth, which is not under volitional control [21]. VR develops in the frontal plane when the head is directed forward and in the sagittal plane when the head is rotated toward one side owing to the interaction between vestibular and neck proprioceptive information controlled by the cerebellum. Highs exhibited the same amplitude of the VR earliest component during both the actual and imagined rotated posture of the head [21]. In line with this behavioural finding, topological EEG studies revealed similar topological asset during actual and imagined rotated posture of the head in highs [14]. The same studies suggested differences in the modes of the cortical elaboration of sensory and imaginative stimuli. During imagery tasks, in fact, highs showed slight, distributed cortical topological changes which were almost not detectable through spectral analysis, whereas lows exhibited task-related localized changes readily detectable through spectral analysis [14,41]. 

The highs’ stronger FE between imagined and actual action [14] together with the greater excitability of their motor cortex [15,16] can increase the likelihood of ideomotor responses, thus reducing the perception of effort and agency. The experience of involuntariness in suggested action, in fact, is one of the most important characteristics of hypnotic behaviour [42] and has been interpreted according to both dissociative [43] and socio-cognitive views [44]. These two main theories can be theoretically reconciled, however, based on the complex nature of movement, which is often automatic and perceived as involuntary also in the ordinary state of consciousness [45]. 

The same EEG study [14] revealed greater topological homogeneity among highs than among lows, during all conditions. Work in progress (Lucas et al., personal communication) confirms this finding during baseline and extends this finding to hypnosis. This may seem to contrast with the observation of different types of highs according to the quality of the scales items they pass rather than according to the scales total score [11]. A possible interpretation is that the processes that are less general than those reflected by topology at mesoscopic level are not detected by the performed topological measures.

### 2.3. Motor Cortex Excitability

Studies of the motor cortex excitability have been performed through transcranial magnetic stimulation (TMS) of the motor cortex and recording of the evoked muscle activity in one hand. They have shown greater excitability of the right motor cortex in highs than in lows in resting conditions and during imagery of movement of the left hand, with mediums exhibiting intermediate excitability [15]. The electromyographic activity at rest and during imagery of movement of the left hand, in fact, showed lower thresholds and higher amplitudes in highs than in lows. In contrast, TMS of the left motor cortex increases the excitability of the motor cortex and decreases the motor threshold in the right hand only during imagery. The high dopaminergic tone [46] of the highs’ cerebral cortex cannot account for the difference in the excitability of the right motor cortex by itself. In contrast to the left motor cortex, which is influenced only by dopamine content, the right cortex is influenced, in fact, also by the reduced cerebellar inhibition possibly due to the reduced volume of the left cerebellar lobules IV and V. The higher excitability of the right motor cortex might take part in the greater proneness of highs to respond to ideomotor suggestions by the left hand. For instance, the larger lowering of the left arm with respect to the right arm during suggestions of arm heaviness could be at least partially accounted for by greater excitability of the right motor cortex [3]. 

### 2.4. Attention, Pain Control and the Cerebellum

The highs’ attention is greatly stable, in fact they are scarcely distractible from their current focus of attention [47,48]. In the general population, low distractibility is associated with high cortical dopamine content [49], and the same seems to occur in highs [50]. The genetic argument supporting their cortical larger dopamine content. However, the polymorphism of the Catechol-O-Methil-Transferase (COMT), responsible for reduced dopamine/noradrenaline catabolism—is weak. Genetic results on hypnotizability-related difference in COMT polymorphism, in fact, are inconsistent among each other, as COMT differences between highs and lows have been found present [51], absent [26,52], present only among males [53], only in participants with peculiar attentional capabilities independently from hypnotizability [54]. Moreover, the studies of frontal functions are somehow inconsistent, as at executive control highs have not been found better than lows (for review, see [7,55]), despite the functional connectivity between the anterior cingulate and the dorsolateral prefrontal cortex is stronger in highs than in lows [56]. It is likely that several factors sustain/modulate the hypnotizability-related attentional characteristics. Thus, it may be worthwhile to note that the cerebellum is involved in both motor and non-motor functions [57] and, specifically, it contributes to the quick changes in the focus of attention. The same cerebellar peculiarities influencing the highs’ sensorimotor behavior can affect their cognitive performance [57]. 

The highs’ cerebellar morphological peculiarities could be also involved in their paradoxical pain control [58]. The highs’ increase in the reported pain intensity and in the amplitude of cortically evoked nociceptive potentials observed after bilateral cerebellar anodal transcranial direct current stimulation (tDCS) in medium-to high hypnotizable participants [58] contrasts, in fact, with the findings obtained in the general population [59] and in low-to-medium hypnotizables [58]. It could be due to reduced inhibition of the regions of the pain matrix involved in cognition and emotion by the cerebellar left lobule VI [60]. Since the motor cortex is involved in the cerebellar induced pain reduction [61], and the highs motor cortex is more excitable than lows’ [15], the highs’ paradoxical behavior should be attributed to the cerebellar projections to other regions of the pain matrix. The insular projections to the prefrontal cortex are good candidates [62,63].

### 2.5. Interoception

Interoception is the perception of the bodily state [64]. It is sustained by afferent signals, central integration, and mental representation of visceral signals [65] and is extremely important, being linked to phenomenal consciousness, body awareness, cognition and affect [66]. The insula is the structure most involved in interoception. Specifically, its anterior and posterior division are mainly connected to the prefrontal and orbitofrontal cortex. Three dimensions of interoception are usually considered: accuracy (IA)—the ability to detect interoceptive signals-, sensitivity (IS)—the interpretation of signals, and awareness, which represents the correspondence between interoceptive accuracy and sensitivity. IA is measured by behavioral tasks, for instance the heartbeats count and its comparison with the ECG recorded heartbeats. IS is measured by questionnaires such as the Multisensory assessment of Interoceptive Awareness (MAIA, [67]), whose subscales indicate the awareness of body sensations (noticing), and that certain body sensations are the sensory aspect of emotional state; (emotional awareness), the tendency to ignore/distract oneself from sensations of pain/discomfort (not distracting), to not experience emotional distress or worry with sensations of pain or discomfort (not worrying), the ability to sustain and control attention to body sensation (attention regulation), to regulate psychological distress by attention to body sensations (self-regulation), to actively listen to the body for insight (body listening) and to experience of one’s body as safe and trustworthy (trusting). The Body Perception Questionnaire [68] refers to the awareness of bodily signals and to the detection of supra and subdiaphragmatic information, thus showing a closer relation with the activity of the autonomic system. 

The difference in the insula grey matter volume and in its connections can be involved in hypnotizability-related differences in interoception. Highs display lower interoceptive accuracy—the ability to detect visceral signals measured by the heartbeat count test—than lows, with mediums exhibiting intermediate values [24]. Accordingly, their heartbeat-evoked cortical potential is smaller than lows’ in the centro-parietal regions, which are reached by projections from the anterior insula [23]. 

The highs’ ability to modify the experience of their body, usually indicated as an effect of dissociation, can be sustained by their low interoceptive accuracy possibly depending on the insula morpho-functional characteristics, which could allow them to feel a body condition different from the real one (and facilitate dissociative experiences). 

### 2.6. Hypnotizability and Brain Injuries

A few hypnotizability-related brain functions allow to hypothesize that highs could be less vulnerable to brain injuries and more resilient to them compared to lows. The topologically different cortical elaboration of sensory and cognitive information—distributed in highs, localized in lows [14]—suggests, in fact, that brain lesions could be less impairing in highs than in lows. Clinical studies, however, are required to ascertain whether brain lesions produce less deficits in highs than in lows. 

The highs’ better cerebrovascular reactivity may buffer transient alteration of blood flow and, theoretically, their stronger FE between imagery and perception/action [14], together with the higher excitability of the motor cortex [15,16], makes highs more prone than lows to take advantage from mental training after brain lesions of any origin [69]. The latter finding suggests that hypnotic assessment could predict the outcome of mental training and Brain Computer Interface interventions, which display large variability in the outcome and is ineffective in part of the general population [70]. Studies in progress are aimed at assessing whether training to mental imagery can increase FE in mediums/lows, thus extending the utilization of mental training to larger part of the population. Preliminary findings show that motor imagery training improves the velocity and accuracy of movement and that the improvement lasts at least two weeks after five days of training [71]. TMS and anodal tDCS aimed at improving motor responses are more expensive and time consuming than mental training, and the duration of their effects has not been consistently reported [72]. 

### 2.7. Cardiovascular Control 

The most important vascular difference between highs and lows is in the post-occlusion flow-mediated endothelial function (FMD, Figure 1B). It is defined as the difference in an artery diameter measured after and before the artery occlusion and is usually tested in the brachial artery. In the general healthy population and in lows, after occlusion the flow-mediated dilation is larger than before it, as the swirling blood flow following dis-occlusion promotes the release of NO from endothelial cells. In highs, the brachial artery post-occlusion flow-mediated dilation is significantly less reduced than in lows during tonic nociceptive stimulation [18] and not reduced at all during mental computation [17,18]. Since FMD is considered a reliable index of cardiovascular health [73], in the absence of risk factors, high hypnotizability could promote a better cardiovascular health.

Endothelial NO controls vascular dilation, thus, is relevant to the function of all organs and systems [74]. Larger post occlusion flow-mediated dilation, in fact, is considered a predictor of less vulnerability to cardiovascular events [75], and drugs containing NO donors are administered in clinical trials [76]. Moreover, NO exerts a central inhibition of the sympathetic activity [77], which is increased in patients with heart failure [78]. In this respect, high hypnotizability may have a favorable prognostic role in case of cardiovascular events. In fact, NO inhalation positively influences the course of cardio-cerebrovascular diseases [79]. We can also hypothesize that highs are less vulnerable to vascular-based cognitive decline owing to their cerebrovascular reactivity [80,81]. Nonetheless, eccessive NO has been associated with Alzheimer earlier degeneration [82]. Thus, it is important to remark that hypnotizability-related differences in basal blood flow have not been observed.

During long lasting relaxation, highs increase their parasympathetic tone more than lows [83]. The parasympathetic tone is indicated by the High Frequency component of the tacogram power spectrum (a signal obtained by reporting the sequence of the distances between consecutive R waves of the ECG). Also, at variance with lows, in standing position the highs’ increase in the Low Frequency component of heart rate variability—related to the sympathetic activity—is not significant [84]. Both findings could be accounted for by higher release of NO in the bulbar regions responsible for sympathetic inhibition in the general population [77]. Thus, the highs’ greater proneness to induce relaxation responses [85] could work as a natural protection against stress.

Low sympathetic activity could also induce more efficient activity of the immune system, with useful effects on autoimmune conditions [86]. Hypnotic treatments, which induces relaxation responses, influence the immune system by modulation of the autonomic activity and consequent greater decreases in highs than in lows in the activity of Natural Killers lymphocytes and lymphocyte proliferative response [87,88]. In highs, hypnotic suggestions of relaxation and wellbeing buffer the decline in NKP, CD8, and CD8/CD4 ratio occurring during examination-related stress in students and upregulate the expression of immune-related genes [89]. In cancer patients and geriatric patients’ positive immune effects of hypnotic relaxation have also been reported [69]. Finally, the highs’ ability to modulate their autonomic activity [83,90,91], challenged by only one study [92] could positively influence their microbiota, whose alteration is involved in cognitive decline owing to the cerebral effects of locally produced cytokines and the activation of afferent vagal fibers [93]. The negative effects of microbiota alteration of have been described, in fact, in case of degenerative physical and cognitive decline [94,95,96,97,98,99,100].

### 2.8. Polymorphism of µ1 Receptors

In the clinical field the large variability of the pain patients’ response to opiates is widely known. In this respect, we studied the polymorphism of µ1 receptors in healthy individuals with different hypnotizability. The results obtained in highs, lows, and controls (represented by anonymous umbilical cords belonging to the general population), was that highs display the less responsive µ1 polymorphism significantly more frequently than lows, with mediums exhibiting intermediate values [101]. Such polymorphism has been associated, in fact, with larger opiates consumption for chronic, post-surgical, and cancer pain [101].

## 3. Limitations and Conclusions

A limitation of the reported studies is that mediums, who represent 70% of the general population, have been seldom enrolled. Thus, now, only part of the hypnotizability-related findings can be extended to the general population [10]. In the studies of FE [14,100] and FMD [17,18] only highs and lows have been recruited. In a few studies—motor cortex excitability [15], interoceptive accuracy [24]—mediums exhibit intermediate values, not always significantly different from highs and lows. In other experiments—cerebellar tDCS stimulation before nociceptive stimulation [58], and cerebral blood flow [19,20], interoception accuracy [24]—the participants have been divided in low-to-medium hypnotizables (according to the score 0–5 of the Italian version of Stanford Hypnotic Susceptibility Scale, form A, [102]) and medium-to-high hypnotizables (score 7–12 on SHSS, A), thus reducing the sensitivity of the study to hypnotizability. 

Finally, some of the suggested mechanisms of the highs’ higher parasympathetic tone, i.e., the sympathetic inhibition by NO at bulbar level should be experimentally confirmed in humans. A question arising from the present approach to hypnotizability is when and how we will be able to perform hypnotic assessment instrumentally. Attempts have been carried out through EEG studies, but they are not satisfying because the suggested indices have been obtained during sessions including suggestions and/or hypnosis [103,104,105]. A possible discriminant index obtained in resting conditions is the Determinism of the EEG Recurrence Plot, which approximates a good separation between highs and lows [106]. A very recent attempt to identify highs and lows during baseline conditions through EEG analysis has been based on the amount of periodic and aperiodic components of the EEG signal [107]. However, to date, only standard scales allow for hypnotic assessment (although their reliability is debated) [2,11].

In conclusion, the present review describes hypnotizability-related physiological characteristics (Figure 2) possibly accounting for a few hypnotic behaviours (i.e., the response to ideomotor suggestions owing to stronger FE between imagery and perception/action and greater excitability of the motor cortex). Such stronger FE may predict better outcome of imagery training in neurological patients. Moreover, the highs’ mode of information processing could sustain greater resilience to brain injuries owing to their distributed processing mode [14]. Their more adaptive cardio- and cerebrovascular functions (availability and sensitivity to endothelial NO [17,18,19,20]) might predict lower vulnerability to vascular events. Finally, the different sensitivity of highs and lows’ µ1 receptor [101] should allow to personalize pharmacological pain therapies. We are aware that several addressed points should be replicated in healthy participants, but the relevance of the presented evidence and advanced hypotheses must be verified by clinical studies. Thus, this review is a call to medical doctors to consider the relevance of hypnotic assessment to their clinical practice.

## Figures and Tables

**Figure 1 brainsci-13-01632-f001:**
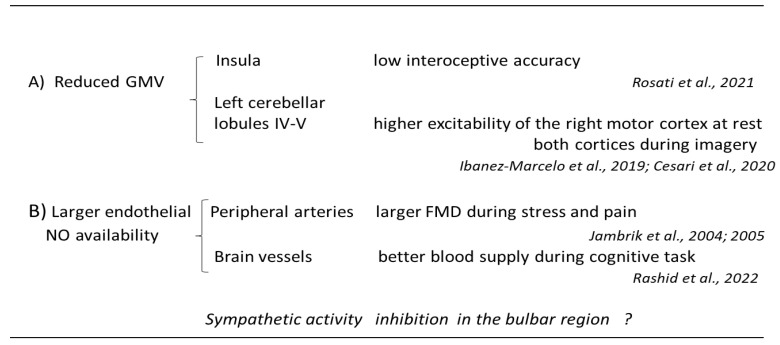
Association between hypnotizability-related morpho-functional differences and behaviour. (**A**) [14,16,24], at brain level; (**B**) at vascular level [17,18,20]. FMD, flow-mediated dilation GMV, gray matter volume; NO, nitric oxide.

**Figure 2 brainsci-13-01632-f002:**
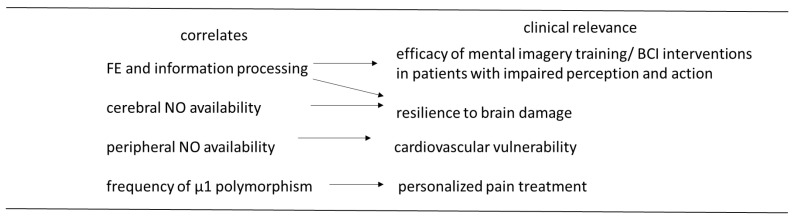
Hypnotizability-related correlates possibly relevant to medicine. FE, functional equivalence between actual and imagined perception/action; NO, nitric oxide.

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
