# Peer review of "Physiological Correlates of Hypnotizability: Hypnotic Behaviour and Prognostic Role in Medicine"

_brainsci, 2023, doi:10.3390/brainsci13121632_

Round 1

Reviewer 1 Report

Comments and Suggestions for Authors

This paper presents a review and summary of the extensive body of recent psychophysiological research in relation to the important trait of hypnotisability, most of it conducted within the last decade.

Whereas the major focus in the neuroscience of hypnosis has been on the use of brain imaging (PET and fMRI) and EEG to investigate response to specific hypnotic suggestions the focus of the research reviewed here involves the use of sophisticated technical developments in the wider field of psychophysiology,  and in particular on the correlates of the stable trait hypnotizability that underlies the capacity to respond to hypnotic suggestions and perhaps other clinically relevant forms of suggestion such as placebo and nocebo.

Familiarity with this body of work is important for clinicians, for whom suggestion and suggestibility may be important factors, experimentalists, who will be responsible to replicate and extend the reported findings, and theorists seeking to build comprehensive models of hypnosis, hypnotisability and suggestion.

The authors are sensitive to the need for replication and are careful to note failures of replication where this has occurred.

While the level of writing and of English (not the authors’ first language) is overall very good there are a few places in which I suspect the English text may be modified slightly to more clearly express the authors’ intentions. I shall indicate below where I perceive this occurs and suggest an alternative wording. The authors should consider if they wish to adopt those minor changes and perhaps in places they may prefer not to. Otherwise there are a couple of apparently clumsy sentences but the meaning is clear enough and I have left those alone.

Subject to this consideration I recommend publication and consider this valuable contribution to the future development of the field.

Line 34 delete “just roughly” and insert “also”. Insert “such” before “as” to read “such as placebo”.

Line 39 and elsewhere please avoid 1 sentence paragraphs. This also occurs in the sentence immediately below and on lines 252 and 345..

Line 40 insert “can be used to” before “classify”.

Line 43 delete “The” and start sentence from “Hypnotizability”.

Line 45 delete “the” and replace “Among” with “Amongst”.

Line 52 insert “and” before “polymorphism”.

Line 87 delete “prisms” and insert after “application” the text “of prisms”.

Line 92 replace “prisms” with “the”; insert “of prisms” after “removal”.

Line 95 delete “The” and instead begin the sentence with “Hypnotizability”.

Line 104 The word “asset” reads oddly here. Please clarify this usage or remove it.

Line 106 remove “content” and replace with “levels”.

Line 130 insert “readily” after “changes”.

Line 131 delete “also”.

Page 6 Figure 1

Although they are defined in the text consider defining acronyms in the note to Figure 1. This is done in Figure 2. Also in naming the table “A,” should probably be “A)”.

Comments on the Quality of English Language

Included previously

Author Response

We thank the reviewer foro is ecouraging and useful comments

This paper presents a review and summary of the extensive body of recent psychophysiological research in relation to the important trait of hypnotisability, most of it conducted within the last decade.

Whereas the major focus in the neuroscience of hypnosis has been on the use of brain imaging (PET and fMRI) and EEG to investigate response to specific hypnotic suggestions the focus of the research reviewed here involves the use of sophisticated technical developments in the wider field of psychophysiology,  and in particular on the correlates of the stable trait hypnotizability that underlies the capacity to respond to hypnotic suggestions and perhaps other clinically relevant forms of suggestion such as placebo and nocebo.

Familiarity with this body of work is important for clinicians, for whom suggestion and suggestibility may be important factors, experimentalists, who will be responsible to replicate and extend the reported findings, and theorists seeking to build comprehensive models of hypnosis, hypnotisability and suggestion.

The authors are sensitive to the need for replication and are careful to note failures of replication where this has occurred.While the level of writing and of English (not the authors’ first language) is overall very good there are a few places in which I suspect the English text may be modified slightly to more clearly express the authors’ intentions. I shall indicate below where I perceive this occurs and suggest an alternative wording. The authors should consider if they wish to adopt those minor changes and perhaps in places they may prefer not to. Otherwise there are a couple of apparently clumsy sentences but the meaning is clear enough and I have left those alone.Subject to this consideration I recommend publication and consider this valuable contribution to the future development of the field.

The indicated lines refer to the original version of the manuscript

Line 34 delete “just roughly” and insert “also”. Insert “such” before “as” to read “such as placebo”.

  • We amended accordingly.

Line 39 and elsewhere please avoid 1 sentence paragraphs. This also occurs in the sentence immediately below and on lines 252 and 345.

  • We expanded the paragraphs by merging more sentences together.

Line 40 insert “can be used to” before “classify”.

  • We amended accordingly.

Line 43 delete “The” and start sentence from “Hypnotizability”.

  • We amended accordingly.

Line 45 delete “the” and replace “Among” with “Amongst”.

  • We amended accordingly.

Line 52 insert “and” before “polymorphism”.

  • We amended accordingly.

Line 87 delete “prisms” and insert after “application” the text “of prisms”.

  • We amended accordingly.

Line 92 replace “prisms” with “the”; insert “of prisms” after “removal”.

  • We amended accordingly.

Line 95 delete “The” and instead begin the sentence with “Hypnotizability”.

  • We amended accordingly.

Line 104 The word “asset” reads oddly here. Please clarify this usage or remove it.

  • We edited the sentence with a more appropriate and specific term.

Line 106 remove “content” and replace with “levels”.

  • We amended accordingly.

Line 130 insert “readily” after “changes”.

  • We amended accordingly.

Line 131 delete “also”.

  • We amended accordingly.

Page 6 Figure 1

Although they are defined in the text consider defining acronyms in the note to Figure 1. This is done in Figure 2. Also in naming the table “A,” should probably be “A)”.

  • We specified the acronyms in the note to Figure 1, too.

Reviewer 2 Report

Comments and Suggestions for Authors

Below are the main requests for authors:

Line 27: “the proneness to enter hypnosis and/or to experience alteration”. I would suggest to do not use the verb “enter”. An alternative could be “..to experience hypnosis and alteration of..”

Line 65: “able to experience deeper hypnotic trance”. The concepts of trance and hypnotic layers are not supported by scientific literature: please, remove deeper and trance.

Line 71: “In highs other studies revealed reduced grey matter volume (GMV) in the insula, larger..”. I suggest to rephrase the sentence, for example “ ..in the insula of highs, as well as…” or  “other investigations on highs revealed..”

Line 146: “the processes less general than”. There is probably a typo here. Do the authors mean “They process” or “the processes are less..”?

Lines 188- 194: This section looks like a cerebellar-centered oversimplification of complex and multifactorial human processes. In fact, emotional perception and expression rely on different neurophysiological (e.g., limbic and cortical regions) and psychological factors (e.g., personality, attachment style etc.). Moreover, previous literature investigated the role of emotions in hypnosis, as well as the role of some hypnotizability correlates (e.g. the attachment style, see Varga) on the human relations (and the tendency to avoid unpleasant emotions). I think the hypnotizability-emotions relationships should be deepened, and much more literature need to be reported. Alternatively, it would be better to eliminate these lines and the word "emotions" from the section. In general, my opinion is that all the paragraph focuses on the cerebellar studies instead of reviewing the literature in the topic. In fact, also the attention is only briefly discussed, and mainly for its possible connections with cerebellum. However, there is a lot of literature investigating the attention-hypnotizability relationship by focusing for example on the cognitive flexibility, prefrontal cortex, brain connectivity etc.   

Line 202: “The highs’ higher excitability of both sides motor areas” Rephrase as follows: “..of the motor areas of both hemispheres”

Section 2.5: the first half of the section is a repetition of the studies already discussed on cerebellum and hypnosis. The second half of the section does not really support the role of the cerebellum on pain regulation in hypnosis. Again, there would be much more literature to discuss in the topic, and other brain regions are more involved than cerebellum. A suggestion might be to bring together sections 2.4 and 2.5 to explicitly focus on the role of the cerebellum in...

Line 251: “A few hypnotizability-related brain functions allow to hypothesize that highs could be less vulnerable to brain injuries”. Do the authors mean, for example, that a hypnotizability-mediated impact is conceivable for a traumatic brain injury or any neurological disease? The sentence is unclear in this form and should be clarified; otherwise, it seems like speculation. On the contrary, the impact of FE on BCI and training is clearer: I would suggest reformulating the title accordingly.

Figure 1. Cerebral aspects: as discussed before, the contribution of many brain areas (PFC, occipital areas etc.) is missing (see eg the review by Landry 2017). I would suggest to remove or modify accordingly the figure. For the same reason, the title sentence “neural mechanisms of hypnosis” is not fully coherent with the content of the paper. The title could be modified in “Physiological correlates of hypnotizability and prognostic role in medicine” or similar. Alternatively, the cerebellum should be mentioned.

Line 287: “high hypnotizability should predict better cardiovascular health”. Is there data supporting this predictive role? it would probably be more prudent to write "could favour a lower risk of..”  or “could promote a better..”

Author Response

We thabk the reviewers for her/his useful co0mments

The indicated lines refer to the original version of the manuscript

Below are the main requests for authors:

 Line 27: “the proneness to enter hypnosis and/or to experience alteration”. I would suggest to do not use the verb “enter”. An alternative could be “..to experience hypnosis and alteration of..”

  • We amended accordingly.

Line 65: “able to experience deeper hypnotic trance”. The concepts of trance and hypnotic layers are not supported by scientific literature: please, remove deeper and trance.

  • We amended accordingly.

Line 71: “In highs other studies revealed reduced grey matter volume (GMV) in the insula, larger..”. I suggest to rephrase the sentence, for example “ ..in the insula of highs, as well as…” or  “other investigations on highs revealed..”

  • We amended accordingly.

Line 146: “the processes less general than”. There is probably a typo here. Do the authors mean “They process” or “the processes are less..”?

  • We better clarified the meaning of this sentence.

Lines 188- 194: This section looks like a cerebellar-centered oversimplification of complex and multifactorial human processes. In fact, emotional perception and expression rely on different neurophysiological (e.g., limbic and cortical regions) and psychological factors (e.g., personality, attachment style etc.). Moreover, previous literature investigated the role of emotions in hypnosis, as well as the role of some hypnotizability correlates (e.g. the attachment style, see Varga) on the human relations (and the tendency to avoid unpleasant emotions). I think the hypnotizability-emotions relationships should be deepened, and much more literature need to be reported. Alternatively, it would be better to eliminate these lines and the word "emotions" from the section. In general, my opinion is that all the paragraph focuses on the cerebellar studies instead of reviewing the literature in the topic. In fact, also the attention is only briefly discussed, and mainly for its possible connections with cerebellum. However, there is a lot of literature investigating the attention-hypnotizability relationship by focusing for example on the cognitive flexibility, prefrontal cortex, brain connectivity etc.   

  • The aim of this section was not to describe the physiology of attention, but only indicate the possible role of the cerebellum in the highs’ attentional characteristics. The text has been modified and a few lines have been added to better explain our intent

Line 202: “The highs’ higher excitability of both sides motor areas” Rephrase as follows: “..of the motor areas of both hemispheres”

  • We amended accordingly.

Section 2.5: the first half of the section is a repetition of the studies already discussed on cerebellum and hypnosis. The second half of the section does not really support the role of the cerebellum on pain regulation in hypnosis. Again, there would be much more literature to discuss in the topic, and other brain regions are more involved than cerebellum. A suggestion might be to bring together sections 2.4 and 2.5 to explicitly focus on the role of the cerebellum in...

  • We have accepted this suggestion and thank the reviewer

Line 251: “A few hypnotizability-related brain functions allow to hypothesize that highs could be less vulnerable to brain injuries”. Do the authors mean, for example, that a hypnotizability-mediated impact is conceivable for a traumatic brain injury or any neurological disease? The sentence is unclear in this form and should be clarified; otherwise, it seems like speculation. On the contrary, the impact of FE on BCI and training is clearer: I would suggest reformulating the title accordingly.

  • The sentence has been clarified in the following lines of the original version: However, we have added a line in the earliest lines of the paragraph: “ It is likely, in fact, that focal lesions may impair distributed activities less than focused networks”

Figure 1. Cerebral aspects: as discussed before, the contribution of many brain areas (PFC, occipital areas etc.) is missing (see eg the review by Landry 2017). I would suggest to remove or modify accordingly the figure. For the same reason, the title sentence “neural mechanisms of hypnosis” is not fully coherent with the content of the paper. The title could be modified in “Physiological correlates of hypnotizability and prognostic role in medicine” or similar. Alternatively, the cerebellum should be mentioned.

  • We accepted the suggestions to modify the title of the paper ( although the paragraph dealing with FE suggests a role of hypnotizability in the ideomotor imagery/feelings of involuntariness/ automaticity, thus with neural mechanisms of hypnosis), and of the figure.

Line 287: “high hypnotizability should predict better cardiovascular health”. Is there data supporting this predictive role? it would probably be more prudent to write "could favour a lower risk of..”  or “could promote a better..”

  • We amended accordingly.

Reviewer 3 Report

Comments and Suggestions for Authors

This paper aims to present a review of the literature pertaining to the physiological correlates of hypnotizability. The goal is important, and the contents of the manuscript are potentially useful for the readers. Specifically, I find the collection of the relevant papers to this topic handy, and worth the effort. Nevertheless, I find the manuscript to rely on far too many unsupported conjectures. This makes the paper potentially misleading to a reader who does not take the time to read into all of the referenced papers. I suggest that the authors revise the manuscript in a way that it is made clear when is the support for an effect robust, and when are the effects only based on hints from one or two exploratory studies. Since most of the effects the authors mention fall into the latter category (partially due to the nature of the field itself), the tone of the conclusions the authors draw need to be significantly mildened, making it clear that they are very tentative. Below are a few more specific comments that my help in revising the manuscript.

-          As the authors point out, hypnotizability (suggestibility during hypnosis) is barely different from waking suggestibility. Furthermore, the manuscript is about physiological correlates of trait-like hypnotizability. It would seem more parsimonistic to talk about suggestibility.

-          The authors write multiple in the manuscript that the highs’ attention is greatly stable (they are less distractible from focus of attention). However, they do not cite evidence for this statement. One of the papers they do cite for one of these sentences is (Presciuttini et al., 2014), but that paper is about COMT polymorphisms, not about attentional stability, focus, or distractibility. The authors also say related to this topic: „The highs’ higher dopaminergic tone, however, is suggested by neuro-psychological tests, such as Stroop test, Vigilance task, Letter fluency task, Choice reaction times (Kallio et al., 2001)”. However, Kallio’s results did not show any significant differences on the mentioned tests between highs and lows. So I don’t find it surprising that the COMT findings are contradictory in the literature as well.

-          „The increase in the reported pain intensity and in the amplitude of cortically evoked nociceptive potentials observed after bilateral cerebellar anodal transcranial direct current stimulation (tDCS) in medium-to high hypnotizable participants (Bocci et al., 2017) contrasts with the findings obtained in the general population and in low-to-medium hypnotizables.” – A citation is missing here for the „findings obtained in the general population and in low-to-medium hypnotizables”.

-          I find the „2.7. Hypnotizability and brain injuries” section lacking grounding in evidence. The authors indicate that „A few hypnotizability-related brain functions allow to hypothesize that highs could be less vulnerable to brain injuries and more resilient to them compared to lows.”, and then the whole section is just conjecture based on indirect findings. For me, this section would make sense more in the discussion than in the main review section.

-          “Hypnotic treatments, which induces relaxation responses, influence the immune system by modulation of the autonomic activity and consequent greater decreases in highs than in lows in the activity of Natural Killers lymphocytes and lymphocyte proliferative response. In highs, hypnotic suggestions of relaxation and wellbeing buffer the decline in NKP, CD8, and CD8/CD4 ratio occurring during examination-related stress in students, and upregulate the expression of immune-related genes.“ – These two sentences are non-trivial, they need citations to back them up.

-          „Finally, the highs’ ability to modulate their autonomic activity (Sebastiani et al., 2005) could positively influence their microbiota (De Benedittis, 2022), whose alteration is also involved in cognitive decline owing to the cerebral effects of locally produced cytokines and the activation af afferent vagal fibers (Weber et al., 2023).” – I find this statement confusing, and potentially misleading. The Sebastiani study only involved Highs, so we do not know whether this “ability” is unique to highs, or would be there for lows as well. Also, the De Benedettis paper only refers to a single empirical test of the hypothesis of hypnotic alteration of the microbiome, and it was a failed attempt with no significant results on the primary measures. So suggesting from these two studies that highs might be protected from cognitive decline is pretty bold.

-          “A possible discriminant index obtained in resting conditions is the Determinism of the EEG Recurrence Plot, which approximates a good separation between highs and lows (Madeo et al., 2013; Chiarucci et al., 2014).” – These findings are far from conclusive, more like first exploratory attempts at tackling this problem. The fact is that currently there is no well established and robust physiological measure or indicator of high hypnotizability, the only current way of testing is to apply suggestions and measure responses to suggestions. It would be good to clarify this in the manuscript.

-          I think that the conclusions of the paper are overreaching the findings. As mentioned above, I think the following statement should be revised to reflect the tentative nature of these claims: “Other correlates of hypnotizability predict c) greater resilience to brain injuries and efficacy of mental training and better cardio- and cerebrovascular functions (availability and sensitivity to endothelial nitric oxide) and allow d) to personalize pharmacological pain therapies (owing to different sensitivity of highs and lows’ μ1 receptors).”, and also since there is no good evidence presented for “b) the stability of attention” among highs, that statement should be revised or deleted as well. Similarly, the final sentence seems to lack justification by the review, so I would suggest that it should be replaced: “Finally, this review is a call to medical doctors to consider the relevance of hypnotic assessment to their clinical practice.”

Comments on the Quality of English Language

The English is good overall, although some mild editing might help readibility.

Author Response

We thank the reviewer for his useful comments

The indicated lines refer to the original version of the manuscript

This paper aims to present a review of the literature pertaining to the physiological correlates of hypnotizability. The goal is important, and the contents of the manuscript are potentially useful for the readers. Specifically, I find the collection of the relevant papers to this topic handy, and worth the effort. Nevertheless, I find the manuscript to rely on far too many unsupported conjectures. This makes the paper potentially misleading to a reader who does not take the time to read into all of the referenced papers. I suggest that the authors revise the manuscript in a way that it is made clear when is the support for an effect robust, and when are the effects only based on hints from one or two exploratory studies. Since most of the effects the authors mention fall into the latter category (partially due to the nature of the field itself), the tone of the conclusions the authors draw need to be significantly mildened, making it clear that they are very tentative. Below are a few more specific comments that my help in revising the manuscript.

-          As the authors point out, hypnotizability (suggestibility during hypnosis) is barely different from waking suggestibility. Furthermore, the manuscript is about physiological correlates of trait-like hypnotizability. It would seem more parsimonistic to talk about suggestibility.

-          The authors write multiple in the manuscript that the highs’ attention is greatly stable (they are less distractible from focus of attention). However, they do not cite evidence for this statement. One of the papers they do cite for one of these sentences is (Presciuttini et al., 2014), but that paper is about COMT polymorphisms, not about attentional stability, focus, or distractibility. The authors also say related to this topic: „The highs’ higher dopaminergic tone, however, is suggested by neuro-psychological tests, such as Stroop test, Vigilance task, Letter fluency task, Choice reaction times (Kallio et al., 2001)”. However, Kallio’s results did not show any significant differences on the mentioned tests between highs and lows. So I don’t find it surprising that the COMT findings are contradictory in the literature as well.

  • Our aim was not to discuss the general mechanisms of attention, but only to suggest the possible role of the cerebellum among the others. However, we modified the text

-          „The increase in the reported pain intensity and in the amplitude of cortically evoked nociceptive potentials observed after bilateral cerebellar anodal transcranial direct current stimulation (tDCS) in medium-to high hypnotizable participants (Bocci et al., 2017) contrasts with the findings obtained in the general population and in low-to-medium hypnotizables.” – A citation is missing here for the „findings obtained in the general population and in low-to-medium hypnotizables”.

References related to the general population have been included (Bocci et al., 2015). Mediums/lows is the control group in the cited study (Bocci et al., 2017)

-          I find the „2.7. Hypnotizability and brain injuries” section lacking grounding in evidence. The authors indicate that „A few hypnotizability-related brain functions allow to hypothesize that highs could be less vulnerable to brain injuries and more resilient to them compared to lows.”, and then the whole section is just conjecture based on indirect findings. For me, this section would make sense more in the discussion than in the main review section.

We agree with the reviewer about the placement of hypothesis and of evidence in different parts of the review. Nonetheless, we think that readers may find it easier to have the evidence and the consequent hypotheses together. Thus, we respectfully reject this suggestion.

-          “Hypnotic treatments, which induces relaxation responses, influence the immune system by modulation of the autonomic activity and consequent greater decreases in highs than in lows in the activity of Natural Killers lymphocytes and lymphocyte proliferative response. In highs, hypnotic suggestions of relaxation and wellbeing buffer the decline in NKP, CD8, and CD8/CD4 ratio occurring during examination-related stress in students, and upregulate the expression of immune-related genes.“ – These two sentences are non-trivial, they need citations to back them up.

We thank the reviewer for this comments. We have included the lacking references.

-          „Finally, the highs’ ability to modulate their autonomic activity (Sebastiani et al., 2005; Jorgensen and Zachariae, 1994, de Benedittis et al., 1994;Santarcangelo et al., 1992  ) could positively influence their microbiota (De Benedittis, 2022), whose alteration is also involved in cognitive decline owing to the cerebral effects of locally produced cytokines and the activation af afferent vagal fibers (Weber et al., 2023).” – I find this statement confusing, and potentially misleading. The Sebastiani study only involved Highs, so we do not know whether this “ability” is unique to highs, or would be there for lows as well. Also, the De Benedettis paper only refers to a single empirical test of the hypothesis of hypnotic alteration of the microbiome, and it was a failed attempt with no significant results on the primary measures. So suggesting from these two studies that highs might be protected from cognitive decline is pretty bold.

We agree. Thus, we have included specific references to support the highs’ proneness to change their autonomic state and some references to support the role of microbiota in cognitive function

-          “A possible discriminant index obtained in resting conditions is the Determinism of the EEG Recurrence Plot, which approximates a good separation between highs and lows (Madeo et al., 2013; Chiarucci et al., 2014).” – These findings are far from conclusive, more like first exploratory attempts at tackling this problem. The fact is that currently there is no well established and robust physiological measure or indicator of high hypnotizability, the only current way of testing is to apply suggestions and measure responses to suggestions. It would be good to clarify this in the manuscript.

Clarified.

-          I think that the conclusions of the paper are overreaching the findings. As mentioned above, I think the following statement should be revised to reflect the tentative nature of these claims: “Other correlates of hypnotizability predict c) greater resilience to brain injuries and efficacy of mental training and better cardio- and cerebrovascular functions (availability and sensitivity to endothelial nitric oxide) and allow d) to personalize pharmacological pain therapies (owing to different sensitivity of highs and lows’ μ1 receptors).”, and also since there is no good evidence presented for “b) the stability of attention” among highs, that statement should be revised or deleted as well. Similarly, the final sentence seems to lack justification by the review, so I would suggest that it should be replaced: “Finally, this review is a call to medical doctors to consider the relevance of hypnotic assessment to their clinical practice.”

We have modified this paragraph. In fact, several addressed points should be replicated in healthy participants, but the real relevance of the evidence and hypotheses here reported should be verified  by clinical studies.

Round 2

Reviewer 3 Report

Comments and Suggestions for Authors

I would like to thank the authors for their efforts in responding to my comments. Overall, I still think that the authors could have done a better job at delinieting evidence-based information and unsupported or barely supported conjecture/theory.

The authors state in the final paragraph that "We are aware that several addressed points should be replicated in healthy participants, but the relevance of the presented evidence and advanced hypotheses must be verified by clinical studies." This statement is a good first step, but it is simply not enough. If a person reads this paper, they will most likely conclude that there is evidence supporting that highs are more resistant to brain injury, or that highs have a more robust microbiome, or that highs have better attention, or that hipnotizability can be measured or screened using EEG. In reality, none of these are accepted in scientific consensus currently, and there is no good supporting evidence for these statements.

Below are some further comments that I would like to point out to the authors, which might help to improve the manuscript.

-         I still think that this statement is not supported by evidence: “Finally, the highs’ ability to modulate their autonomic activity [90,91,92] could positively influence their microbiota”. In fact the only study that assessed this so far have returned a negative evidence. I suggest deleting this or rephrasing so that it is clear that this is currently nothing more than a theory, and the evidence so far is contradicting this theory.

-          In response to my previous comment saying that “A possible discriminant index obtained in resting conditions is the Determinism of the EEG Recurrence Plot, which approximates a good separation between highs and lows (Madeo et al., 2013; Chiarucci et al., 2014).” – These findings are far from conclusive, and that these are not reliable measures of suggestibility, the authors did not really respond. The original sentence remained unchanged. I still think it is an unacceptable exaggeration that the Determinism of the EEG Recurrence Plot approximates a good separation between highs and lows. I would like to suggest again to milden the wording here. For example by saying that there are exploratory attempts at finding alternative screnning tools, and one of these, using the Determinism of the EEG Recurrence Plot, have seen some supporting evidence, but the findings are not yet conclusive. Furthermore, the sentence added to the end of this paragraph in this new version seems incomplete: “Thus, only standard scales allow hypnotic assessment, although their reliability is debated (“

Comments on the Quality of English Language

The English language could be improved somewhat, but it is OK overall.

Author Response

I would like to thank the authors for their efforts in responding to my comments. Overall, I still think that the authors could have done a better job at delinieting evidence-based information and unsupported or barely supported conjecture/theory.

The authors state in the final paragraph that "We are aware that several addressed points should be replicated in healthy participants, but the relevance of the presented evidence and advanced hypotheses must be verified by clinical studies." This statement is a good first step, but it is simply not enough. If a person reads this paper, they will most likely conclude that there is evidence supporting that highs are more resistant to brain injury, or that highs have a more robust microbiome, or that highs have better attention, or that hipnotizability can be measured or screened using EEG. In reality, none of these are accepted in scientific consensus currently, and there is no good supporting evidence for these statements.

We understand the reviewer point, but we do not agree. In fact, we have clearly indicated in the introduction that we report facts and hypotheses (From this perspective, we describe facts and the hypotheses which can be suggested based on current evidence.”). Hypotheses are relevant to stimulate the interest of clinicians (neurologists, cardiologists, anesthesists) into clinical studies able to support the advanced hypotheses.

We wish also remark that the role of the physiological correlates of hypnotizability in both the neural mechanisms of hypnosis (response to ideomotor suggetions) and clinical prognosis is being studied only in our lab, thus it is quite difficult to find literature supporting pur hypotheses.

However, in the conclusions we have milded some statements by changing can into may etc etc.

Below are some further comments that I would like to point out to the authors, which might help to improve the manuscript.

-         I still think that this statement is not supported by evidence: “Finally, the highs’ ability to modulate their autonomic activity [90,91,92] could positively influence their microbiota”. In fact the only study that assessed this so far have returned a negative evidence. I suggest deleting this or rephrasing so that it is clear that this is currently nothing more than a theory, and the evidence so far is contradicting this theory.

Highs are known to displace their autonomic valance toward a parasympathetic prevalence during hypnotic relaxation (De Benedittis et al., 1994), during simple relaxation (Santarcangelo et al., 1992; 2010). Only one study did not report such parasympathetic prevalence in highs during simple relaxation (Zachariae et al., 2000).

In the healthy general population relaxation enhances parasympathetic and decreases sympathetic activity (Bonaz et al., 2013).

Since noradrenaline and acetilcholine influence the gut microbiota (Lyte et al., 2011), it is obvious that greater proneness to relax will promote the beneficial effects of parasympathetic activity on microbiota alfa diversity (Park et al., 2022). On the other hand, stress-related alteration of autonomic activity strongly alters microbiota diversity.

We report here part of a related study: “The brain and the gut communicate through the autonomic nervous system and the circumventricular organs both in physiologic and pathologic conditions.” The stress-related pathways associating autonomic control and gut microbiota include: “activation of … and of the sympathetic nervous system, vagus nerve inhibition on inflammatory pathways, …, the intestinal microbiota-brain axis.” (Bonaz et al., 2013, Park et al., 2022)

In conclusion, our hypothesis regarding the lower vulnerability of highs to microbiota alteration is not based on the quoted review by De Benedittis, but on published experimental studies. We did not cite all of them. We report them in the revised version (Bonaz et al., 2013; Lyte et al., 2011).

Brain-gut interactions in IBD: the different actors and pathways through which stress may play a deleterious role in IBD. (From Bonaz et al., 2013).

-          In response to my previous comment saying that “A possible discriminant index obtained in resting conditions is the Determinism of the EEG Recurrence Plot, which approximates a good separation between highs and lows (Madeo et al., 2013; Chiarucci et al., 2014).” – These findings are far from conclusive, and that these are not reliable measures of suggestibility, the authors did not really respond. The original sentence remained unchanged. I still think it is an unacceptable exaggeration that the Determinism of the EEG Recurrence Plot approximates a good separation between highs and lows. I would like to suggest again to milden the wording here. For example by saying that there are exploratory attempts at finding alternative screnning tools, and one of these, using the Determinism of the EEG Recurrence Plot, have seen some supporting evidence, but the findings are not yet conclusive. Furthermore, the sentence added to the end of this paragraph in this new version seems incomplete: “Thus, only standard scales allow hypnotic assessment, although their reliability is debated (“

We thank the reviewer for this request of clarification. However, we think to have clearly written that “a possible index” is the Determinism of Recurrence Quantification Analysis. In fact, an earlier study shows that only one high and one low are placed in the wrong category by this analysis (Madeo et al., 2013, not cited any longer in order to reduce the number of references). Moreover, a successive study (Chiarucci et al., 2014, cited) shows that the results can be improved by further analyses of non-linear systems. Since these are technical aspects, we thought that it was better to omit so many details. In this revision of the paper, we could introduce a very very recent evidence regarding the possibility to identify not hypnotized highs and lows through EEG analysis focused on the periodic and aperiodic components of the EEG signal (Landry et al., 2023).

We added the lacking reference to the indicated sentence.

Added references

Bonaz B, Bernstein CN. Brain-gut interactions in inflammatory bowel disease. Gastroenterology 2013;144:36–49.)

Lyte, M., Vulchanova, L. & Brown, D.R. Stress at the intestinal surface: catecholamines and mucosa–bacteria interactions. Cell Tissue Res 343, 23–32 (2011). https://doi.org/10.1007/s00441-010-1050-0

Landry, M., da Silva Castanheira, J., Sackur, J., Raz, A., Ogez, D., Rainville, P., Jerbi, K. (2023). Unravelling the neural dynamics of hypnotic susceptibility: Aperiodic neural activity as a central feature of hypnosis, bioRchive, pre-print.

Park S, Wu X. Modulation of the Gut Microbiota in Memory Impairment and Alzheimer's Disease via the Inhibition of the Parasympathetic Nervous System. Int J Mol Sci. 2022 Nov 5;23(21):13574. doi: 10.3390/ijms232113574. PMID: 36362360; PMCID: PMC9657043.
